# Utility of REMS-Derived Fragility Score and Trabecular Bone Score in Evaluating Bone Health in Type 2 Diabetes Mellitus

**DOI:** 10.3390/diagnostics15222877

**Published:** 2025-11-13

**Authors:** Antonella Al Refaie, Caterina Mondillo, Guido Cavati, Sara Gonnelli, Maria Dea Tomai Pitinca, Elena Ceccarelli, Paola Pisani, Luigi Gennari, Stefano Gonnelli, Carla Caffarelli

**Affiliations:** 1Section of Internal Medicine, Department of Medicine, Surgery and Neuroscience, University of Siena, 53100 Siena, Italy; antonellaalrefaie@gmail.com (A.A.R.); caterinamondillo28@gmail.com (C.M.); guido.cavati@student.unisi.it (G.C.); saragonnelli@libero.it (S.G.); dea_to79@yahoo.it (M.D.T.P.); ele.ceccarelli77@gmail.com (E.C.); luigi.gennari@unisi.it (L.G.); stefano.gonnelli@unisi.it (S.G.); 2Division of Internal Medicine I, San Giuseppe Hospital, 50053 Empoli, Italy; 3Department of Geriatrics, University Hospital of Nice, 06000 Nice, France; 4Institute of Clinical Physiology, National Research Council, 37100 Lecce, Italy; pisanipaolaifc@gmail.com

**Keywords:** radiofrequency echographic multi-spectrometry (REMS), trabecular bone score (TBS), bone mineral density (BMD), Fragility Score (FS), type 2 diabetes mellitus (T2DM), osteoporosis, major osteoporotic fractures (MOF)

## Abstract

**Background/Objectives**: A significantly higher fracture risk characterizes Type 2 diabetes mellitus (T2DM) patients when compared to the non-diabetic population, even though their average bone mineral density (BMD) tends to be normal or high. This elevated risk is primarily driven by defective bone quality. The trabecular bone score (TBS) and radiofrequency echographic multispectrometry (REMS) have recently been proposed to improve the assessment of bone quality in T2DM individuals. This study aimed to evaluate whether TBS and REMS can improve the identification of osteoporosis and fracture risk in these patients. **Methods**: BMD was measured in 223 consecutive T2DM patients (126 women and 97 man) and 102 controls. BMD values for the lumbar spine (LS), femoral neck (FN), and total hip (TH) were obtained via both dual-energy X-ray absorptiometry (DXA) and radiofrequency echographic multi-spectrometry (REMS). In all patients, TBS and Fragility Score (FS) by REMS were measured and prior major osteoporotic fractures (MOF) were assessed. **Results**: All BMD T-scores measured by REMS were significantly lower than those obtained by DXA at both lumbar and femoral sites. T2DM patients with previous MOF exhibited lower T-scores for both BMD-LS and BMD-TH, as assessed by DXA and REMS, compared with patients without fractures. However, these differences reached statistical significance for BMD-TH with both techniques and for BMD-LS with REMS, but not for BMD-LS with DXA. Moreover, patients with a history of MOF had significantly lower TBS values (*p* < 0.05) and significantly higher FS values at both lumbar (*p* < 0.05) and femoral (*p* < 0.01) sites compared with those without fractures. **Conclusions**: The results of this study suggest that the parameters obtained using REMS technology (BMD and FS) may be valuable tools for improving the diagnosis of osteoporosis and assessing fracture risk in patients with T2DM.

## 1. Introduction

Type 2 Diabetes Mellitus (T2DM) poses an escalating global health challenge. Indeed, the aging population, the adoption of increasingly sedentary lifestyles, and the spread of obesity are crucial factors in the growing prevalence of T2DM globally. Current estimates from the International Diabetes Federation indicate that in 2021, approximately 537 million adults worldwide were living with diabetes, and that number will rise to over 780 million by 2045 [1]. T2DM is associated with a wide range of complications, including cardiovascular disease, chronic kidney disease, retinopathy, and peripheral neuropathy, making it a major driver of disability-adjusted life years (DALYs) and healthcare expenditures.

Numerous scientific studies conducted over the past two decades have established that T2DM is associated with an increased risk of fragility fractures, most notably at the hip but also frequently at the wrist and ankle, and that these fractures exert a substantial impact on both quality of life and DALYs in diabetic patients [2,3,4,5,6]. Evidence further indicates that these fractures frequently occur in patients with normal or even moderately elevated bone mineral density (BMD). This observation highlights that BMD assessment by dual-energy X-ray absorptiometry (DXA) and common fracture risk algorithms substantially underestimate fracture risk in individuals with T2DM, suggesting that skeletal fragility in this population is primarily attributable to compromised bone quality and impaired microarchitecture rather than to reduced bone mass [2,4,7,8]. The causes of bone fragility in T2DM have not been fully elucidated; however, they are attributed to alterations in bone microarchitecture and matrix composition due to reduced bone turnover and the accumulation of advanced glycation end-products (AGEs), which interact with the RAGE/NF-κB signaling pathways, thereby promoting oxidative stress, inflammation, and the progression of diabetic complications [9,10]. This clinical context has stimulated increasing interest in the development of easy-to-use and reliable diagnostic methods for assessing qualitative and structural skeletal alterations, as well as fracture risk, in patients with T2DM [11]. Among these, the most widely adopted tool has been the trabecular bone score (TBS).

TBS is a non-invasive method that utilizes gray-level pixel analysis of DXA images to provide an indirect evaluation of bone microarchitecture by analyzing trabecular texture independently of BMD [12,13,14]. Several large population-based studies in T2DM patients have consistently reported reduced TBS values, despite higher BMD at both the spine and femur compared with non-diabetic individuals [15]. Accordingly, TBS can be considered a simple, non-invasive index of bone microarchitecture that contributes to the evaluation of skeletal fragility in T2DM patients [16]. A major limitation of TBS, however, is its inability to provide structural information on the femoral bone.

Previous research has highlighted the potential of radiofrequency echographic multispectrometry (REMS), an innovative non-ionizing technology, to overcome several inherent limitations of DXA in the assessment of bone health among patients with T2DM [17]. In an exploratory study conducted in a cohort of postmenopausal women with diabetes, DXA-derived BMD values were, as expected, higher in diabetic women compared with non-diabetic controls. By contrast, REMS-derived BMD values were significantly lower in the diabetic group [18].

A recent advancement in the REMS methodology is the development of the Fragility Score (FS), a dimensionless parameter ranging from 0 to 100, obtained by comparing the patient’s spectral profile with reference models derived from fractured and non-fractured individuals [19,20]. The FS provides an index of bone quality independent of BMD. Specifically, FS values represent the percentage of analyzed bone segments whose spectral characteristics more closely resemble those of a fractured rather than a non-fractured model. Accordingly, lower values indicate better bone quality, whereas higher values reflect increased skeletal fragility. Evidence supporting the clinical utility of FS is steadily increasing. Notably, a longitudinal study with up to five years of follow-up conducted by Pisani et al. demonstrated that FS at both the lumbar spine and femur discriminated more effectively between fractured and non-fractured subjects than T-scores obtained by either DXA or REMS [21]. To date, however, no data are available regarding the predictive value of FS for fracture risk in patients with T2DM.

The primary objective of this single-center study was therefore to evaluate REMS-derived parameters (BMD and FS) in a large cohort of male and female patients with T2DM, and to explore their associations with clinical characteristics and the presence of osteoporotic fractures.

## 2. Materials and Methods

### 2.1. Study Population

Three hundred consecutive Caucasian outpatients with Type 2 diabetes mellitus (129 men and 171 women) were recruited between January 2024 and June 2025 from the Diabetes Unit, Department of Internal Medicine, at the University Hospital of Siena. All participants provided written informed consent. The study was approved by the Institutional Review Board of Siena University Hospital (ID-21211/21; approved on 13 December 2021). All data were anonymized prior to statistical analysis. Eligibility criteria included: age between 50 and 80 years; postmenopausal status for women; body mass index (BMI) ranging from 18.5 to 39.9 kg/m^2^; age at T2DM diagnosis >30 years; and glycated hemoglobin (HbA1c) < 7.5%. Patients with a history of anti-osteoporotic treatment (with the exception of calcium or vitamin D supplementation), malignant or metabolic bone diseases (e.g., cancer, multiple myeloma, hyperparathyroidism), or therapies known to affect bone metabolism were excluded. The control group comprised 120 consecutive non-diabetic individuals (53 men and 67 women), referred to the Outpatient Clinic of the same Department during the same study period, in addition to a small number of healthy volunteers recruited among the hospital staff. Inclusion criteria were identical to those applied to the T2DM cohort (age 50–80 years, BMI 18.5–39.9 kg/m^2^, and postmenopausal status for women). Non-diabetic subjects with comorbidities or receiving treatments potentially interfering with bone metabolism were excluded. For all participants, a comprehensive personal and family medical history was obtained, including information on smoking habits, alcohol consumption, menopausal duration, T2DM duration, and concomitant comorbidities. Anthropometric measurements were collected under standardized conditions: height and weight were recorded, and BMI was calculated as weight (kg) divided by height squared (m^2^). A total of 52 individuals (40 with T2DM and 12 controls) were excluded because they did not meet the inclusion criteria or withdrew consent, and 43 (37 with T2DM and 6 controls) were excluded due to poor-quality BMD or REMS scans. The final analysis included 223 participants with T2DM and 102 controls. The flow chart showing the distribution of participants in the study is presented in Figure 1.

### 2.2. Dual-Energy X-Ray Absorptiometry (DXA)

BMD measurements for the lumbar spine (LS), femoral neck (FN), and total hip (TH) were obtained for all subjects via dual-energy X-ray absorptiometry (Discovery W, Hologic, Waltham, MA, USA), following standardized protocols. Diagnostic criteria, based on World Health Organization (WHO) guidelines, classified osteoporosis as a T-score ≤ −2.5 and osteopenia as a T-score ranging from −1.0 to −2.5. Sex-specific Italian reference data were used for T-score calculation. To provide additional information on bone microarchitecture, the trabecular bone score was derived from standard anteroposterior lumbar spine DXA images using TBS iNsight software (version 2.2.0.0, Medimaps SA, Bordeaux, France). TBS values were generated in an operator-independent, fully automated manner.

### 2.3. Radiofrequency Echographic MultiSpectrometry (REMS)

Bone mineral density was also assessed using Radiofrequency Echographic Multispectrometry (EchoStation, Echolight SpA, Lecce, Italy) with a convex transducer operating at 3.5 MHz. The methodology of REMS technology, including precision and reproducibility, has been described in detail elsewhere [17,21]. Briefly, the probe was positioned on the abdomen or hip to visualize the target site, with depth and focus adjusted as needed. Raw signals were processed to generate a patient-specific spectrum, automatically compared with sex-, age-, site-, and BMI-matched reference models from a dedicated database. Skeletal fragility was also evaluated using the REMS-derived Fragility Score (FS), which reflects bone microarchitecture independently of BMD at the lumbar spine and femoral neck. FS ranges from 0 (normal microarchitecture) to 100 (maximum fragility) and is calculated as the proportion of scan lines more consistent with a fragile (fractured) than with a normal bone model. FS has been validated as a predictor of 5-year fragility fracture risk in both women and men [20,21]. Low values indicate preserved microarchitecture and lower fracture risk, whereas high values reflect compromised microarchitecture and increased risk.

### 2.4. Laboratory Tests and Fractures Assessment

After an overnight fast (≥12 h), venous blood samples were collected to measure glycated hemoglobin, creatinine, calcium, phosphate, parathyroid hormone (PTH), and 25-hydroxyvitamin D (25OHD). Biochemical parameters were assessed using a colorimetric method (Autoanalyzer, Falcor 350, Menarini, Florence, Italy). Parathyroid hormone (PTH) concentrations were determined via an immunoradiometric assay (Total Intact PTH, Antibodies Lab Inc., Santee, CA, USA), with intra- and inter-assay CVs of 3.6% and 4.9%. Biochemical Assays Serum 25-hydroxyvitamin D (25OHD) levels were quantified using a chemiluminescence immunoassay (LIAISON 25OHD Total Assay, DiaSorin Inc., Stillwater, MN, USA), exhibiting intra- and inter-assay coefficients of variation (CVs) of 6.8% and 9.2%, respectively.

### 2.5. Fracture Assessment

Prior Major Osteoporotic Fractures (MOF)—including those of the hip, spine, wrist, and humerus—were assessed in the T2DM group via both self-report and subsequent verification through clinical and radiological records.

### 2.6. Statistical Analysis

The values in the study are presented as “mean ± standard deviation (SD). The normality of the distribution of outcome variables was assessed using the Kolmogorov–Smirnov test. Clinical data and initial values of the measured variables in the study groups were compared using the Student’s *t*-test and Mann–Whitney U-test, depending on the appropriateness of the data distribution. Categorical variables were subjected to comparison using the Chi-square test or Fisher’s exact test, as deemed appropriate. Associations between different parameters were examined through Pearson’s correlation or Spearman’s correlation, as appropriate, or via partial correlation analysis. All statistical analyses were performed using the SPSS statistical package for Windows version 16.0 (SPSS Inc., Chicago, IL, USA).

## 3. Results

The T2DM and control groups were matched for age, height, PTH, and 25OHD levels [Table 1]. A significant difference was observed for BMI, which was higher in the T2DM cohort (*p* < 0.01). The mean diabetes duration was 12.8 ± 10.6 years. BMD findings diverged significantly by measurement technique: DXA showed higher BMD across all sites in T2DM patients, with LS-BMD and TH-BMD differences reaching significance (*p* < 0.01). In sharp contrast, REMS recorded lower BMD values at all sites in T2DM patients, with a significant reduction (*p* < 0.05) noted solely for LS-BMD.

As illustrated in Figure 2, a clear and statistically significant disparity was observed between the two measurement modalities. The mean T-scores for BMD determined by REMS were substantially lower than the corresponding DXA T-scores at the lumbar spine (*p* < 0.001) and the total hip (*p* < 0.01).

Figure 3 shows the percentage of T2DM men (A) and T2DM women (B) classified as “osteoporotic”, “osteopenic” or “normal” on the basis of BMD T-score values obtained by DXA and REMS technique, respectively. Regarding the male population, it is evident that the REMS technique allows a greater number of T2DM patients to be classified as osteoporotic and osteopenic than DXA (22.8% and 47.4% vs. 7.0% and 38.6%, respectively). Moreover, classification of bone status showed a marked difference between the two techniques, particularly in the T2DM group. Among T2DM men, the percentage classified as normal was substantially higher by DXA (54.4%) than by REMS (29.8%). The female population exhibited a similar pattern of reclassification. Specifically, REMS classified a significantly greater proportion of T2DM women as osteoporotic (43.9%) compared to DXA (17.8%). Conversely, the percentage of T2DM women categorized as osteopenic or normal was higher when assessed by DXA (46.7% and 35.5%, respectively) compared to REMS (40.7% and 15.4%, respectively).

A history of major osteoporotic fractures was reported in 42 (=18.8%) T2DM patients. More specifically, this was observed in 11 (=11.3%) males and 31 (=22.8%) females. Values of BMD expressed as T-score at lumbar spine (LS) and at total hip (TH) by DXA and REMS technique in T2DM patients with or without MOF are shown in Figure 4. As expected, the T2DM patients with previous MOF presented significantly lower values of T-score both BMD-LS and BMD-TH by DXA and T-score BMD-LS and BMD-TH by REMS with respect to those without fractures; however, the value at the T-score BMD LS by REMS technique, demonstrated higher statistical significance (*p* < 0.01).

Figure 5 illustrates the Trabecular Bone Score, measured by DXA, and the Fragility Score, measured by REMS, in patients with T2DM with or without MOF. As expected, patients with a history of MOF had significantly lower TBS values and significantly higher FS values at both lumbar and femoral sites compared with those without fractures (*p* < 0.05).

## 4. Discussion

In this study, we evaluated for the first time the role of REMS-derived parameters (BMD and FS) in a large cohort of male and female patients with T2DM, by comparing them with TBS and BMD values obtained by DXA. We observed that the REMS method classified as “osteoporotic” twice as many men as women with T2DM. Moreover, FS demonstrated a discriminative capacity comparable to that of TBS in distinguishing between T2DM patients with and without major osteoporotic fractures. A substantial body of evidence indicates that skeletal fragility and fragility fractures, particularly those involving the hip and distal radius, represent frequent and clinically relevant complications in individuals with T2DM [22,23]. This phenomenon has often been referred to as the “diabetic bone paradox”, as affected patients typically exhibit normal or even increased BMD values. The apparent dissociation between BMD and fracture risk in T2DM is largely explained by alterations in bone quality, including microarchitectural deterioration, accumulation of AGEs within collagen, impaired bone remodeling, and changes in bone material properties, ultimately leading to reduced bone strength despite preserved or elevated bone mass [24]. Unfortunately, bone quality is not easily quantifiable; indeed, the most accurate methods for evaluating bone quality and strength—such as bone biopsy and microindentation—are invasive procedures and therefore not routinely employed in clinical practice [11,24,25].

High-resolution peripheral quantitative computed tomography (HR-pQCT) provides three-dimensional bone images of remarkable quality and represents a valuable non-invasive technique for assessing volumetric bone mineral density as well as other structural parameters. However, due to its technical characteristics, HR-pQCT is not suitable for routine clinical use. Moreover, it remains uncertain whether HR-pQCT parameters can reliably predict fracture risk in patients with T2DM [24]. In parallel, considerable interest has been directed toward quantitative ultrasound (QUS) techniques. QUS is particularly attractive because it evaluates bone properties through the attenuation and reflection of ultrasound waves and offers practical advantages over DXA, including low cost, portability, and the absence of ionizing radiation [26]. Nonetheless, studies investigating QUS in patients with T2DM have yielded conflicting results, and QUS has not been shown to differentiate between diabetic patients with and without fractures [27]. In addition, QUS presents important limitations: measurements are restricted to peripheral skeletal sites, and the wide variety of devices currently available—each employing different measurement techniques and parameters—compromises the comparability and reproducibility of results [17,24].

Radiofrequency Echographic Multi-Spectrometry represents an innovative, ultrasound-based technology that enables the quantitative assessment of bone quality and strength without exposure to ionizing radiation [28]. The present study, conducted in a large and well-characterized cohort of men and women with T2DM demonstrated that BMD values obtained by REMS were significantly lower in patients with T2DM compared with age- and sex-matched healthy controls. Furthermore, the proportion of diabetic participants classified as osteoporotic according to REMS-derived BMD was markedly higher than that observed when classification was based on DXA-derived BMD values. These findings are of particular relevance because they appear to contrast with most previous investigations employing DXA in T2DM populations, which have generally reported normal or even slightly elevated BMD values relative to non-diabetic controls [2,3,4,7]. The present study confirms the findings of a preliminary study by Caffarelli et al. conducted on a cohort of postmenopausal women with T2DM. That study reported that DXA-derived BMD values were higher in the diabetic group compared to non-diabetic controls, whereas REMS-derived BMD values were significantly lower in the diabetic cohort. This discrepancy resulted in a marked difference in osteoporosis prevalence, with REMS identifying 47% of patients as osteoporotic, compared to only 28% by DXA [18]. The discrepancy between REMS and DXA measurements may be explained, at least in part, by the ability of REMS to eliminate artifacts commonly observed in patients with T2DM, such as osteophytes, degenerative joint changes, vascular calcifications, and diffuse idiopathic skeletal hyperostosis, which typically lead to an overestimation of BMD by DXA [17,28,29,30]. Thanks to its capacity to recognize and exclude these confounding factors, REMS may provide a more accurate assessment of true skeletal status in this patient population. An additional and clinically meaningful observation from this study is that BMD values measured by REMS were significantly lower in T2DM patients with a documented history of major osteoporotic fractures compared with those without fractures. This association underscores the potential of REMS to capture alterations in bone microarchitecture and strength that are not adequately reflected by DXA-derived BMD measurements [17,28]. The capability of REMS to detect individuals at increased risk of fracture has been extensively documented [28,31,32]. In particular, a longitudinal study involving a large cohort of Caucasian women demonstrated that the REMS T-score represented a reliable and independent predictor of incident fragility fractures over a follow-up period of up to five years [31].

Taken together, these results suggest that REMS may share similarities with the Trabecular Bone Score in its ability to reflect alterations in bone quality among patients with T2DM. Indeed, Shevroja et al. reported that several studies involving more than 40,508 individuals (including 4269 with diabetes) consistently demonstrated that TBS values are lower in diabetic patients compared to controls, whereas BMD values are higher in diabetics than in controls. These findings suggest that TBS may serve as a valuable tool for assessing fracture risk in patients with diabetes [14]. However, in one study, the difference in TBS between diabetic and non-diabetic individuals was statistically significant in women but not in men [33]. Several studies have further evaluated the ability of TBS to discriminate between T2DM patients with fractures and control subjects, showing that TBS values are reduced in diabetic patients and that TBS is associated with fracture risk [34]. In this regard, several studies have confirmed the role of TBS as an instrument that enhances diagnostic accuracy in distinguishing fragility fractures within the context of T2DM-related secondary osteoporosis [13,15,35]. Nevertheless, important limitations restrict its clinical applicability. Since TBS is derived from lumbar spine DXA images, it is inherently subject to the same sources of error and artifacts that affect BMD measurements, including degenerative spinal changes, osteophytes, and vascular calcifications. Moreover, body mass index and abdominal fat are known to influence TBS values, often leading to an underestimation of trabecular quality in overweight or obese individuals, a particularly relevant concern in patients with T2DM, where obesity is highly prevalent [36]. However, the most recent versions of the TBS software seem to be less affected by the regional fat and BMI [36,37]. A further limitation of TBS is that, as it cannot be applied to the femur, it fails to provide adequate information regarding the structural characteristics of cortical bone.

The Fragility Score is a dimensionless REMS-based measure of skeletal fragility that evaluates bone microarchitecture independently of BMD, specifically at the spine and femoral neck [20]. Several investigations have demonstrated that FS exhibits a significant association with both prevalent and incident fragility fractures [20,21]. Another interesting finding of this study is that FS values at both lumbar and femoral sites, were significantly higher in T2DM patients with a history of MOF. This observation appears to confirm previous investigations demonstrating that the FS shows a significant association with both prevalent and incident fragility fractures, independently of BMD values [21,38]. In their study, Pisani et al. investigated a cohort of 1989 Caucasian participants of both sexes, aged between 30 and 90 years, to assess the incidence of fractures over a follow-up period of up to five years. Their findings highlighted the strong predictive value of the FS in identifying individuals at increased risk of incident fragility fractures, with an AUC of 0.811 in women and 0.780 in men. Notably, the ability of the FS to discriminate subjects at risk for fragility fractures was superior to that of REMS-derived BMD T-scores and DXA-derived BMD T-scores [21]. Similarly, Lalli et al. reported, in a cohort of 175 patients with either primary or disuse osteoporosis, that FS values were significantly higher among individuals with a history of fractures than among those without previous fracture events [38]. These findings further support the clinical utility of the FS as a reliable tool for fracture risk stratification, potentially enhancing the accuracy of current assessment methods based solely on densitometric parameters.

An important advantage of FS, especially at the femoral site, lies in its ability to provide clinically relevant information on skeletal fragility in a substantial proportion of diabetic patients with a BMI > 40 Kg/m^2^, in whom TBS cannot be reliably assessed [36]. Therefore, the findings of the present study suggest that REMS may complement and integrate TBS in the assessment of bone status in patients with T2DM.

Our study presents certain limitations. First, its cross-sectional design does not allow for the establishment of causal relationships between the investigated parameters. Second, the sample included a relatively small number of subjects with major fragility fractures, which may limit the generalizability of the findings. Nonetheless, the study also has important strengths. To the best of our knowledge, this is the first investigation to evaluate REMS parameters (FS and BMD) specifically in male patients with diabetes. Moreover, being a single-center study, fracture assessment could be directly based on radiological reports, thereby ensuring methodological consistency.

## 5. Conclusions

In conclusion, the findings of this study support the potential of REMS technology as a valuable tool to improve the diagnosis of osteoporosis and the assessment of fracture risk in patients with type 2 diabetes mellitus.

## Figures and Tables

**Figure 1 diagnostics-15-02877-f001:**
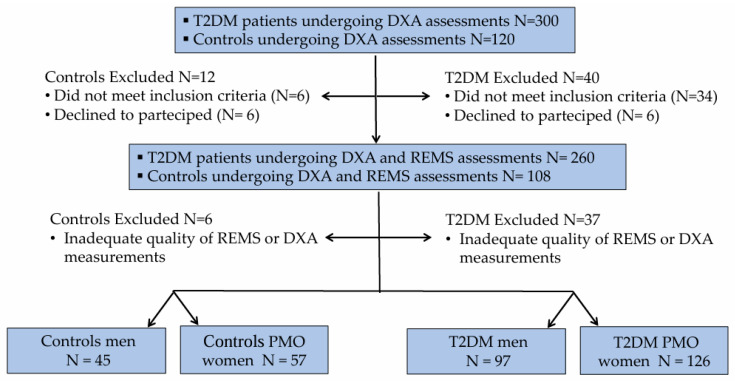
Flow chart showing the distribution of T2DM patients and controls to the study.

**Figure 2 diagnostics-15-02877-f002:**
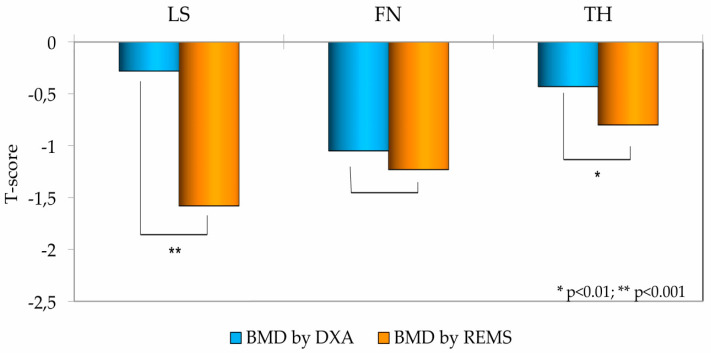
Bone Mineral Density (BMD) T-scores at the Lumbar Spine (LS), Femoral Neck (FN), and Total Hip (TH) in T2DM patients, measured by DXA and REMS technique.

**Figure 3 diagnostics-15-02877-f003:**
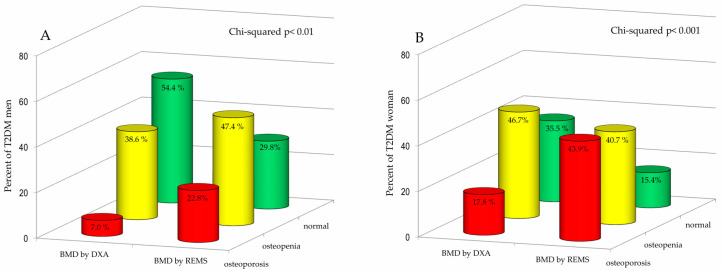
Percentage of Type 2 Diabetes Mellitus (T2DM) men (**A**) and women (**B**) categorized as osteoporotic, osteopenic, or normal based on BMD T-score values obtained by DXA and REMS techniques.

**Figure 4 diagnostics-15-02877-f004:**
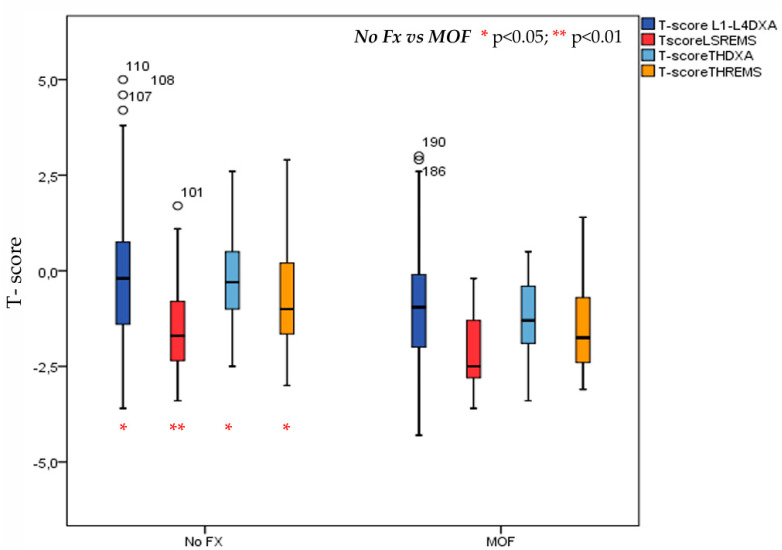
Values of BMD expressed as T-score at lumbar spine (LS) and at total hip (TH) by DXA and REMS technique in T2DM patients with or without MOF.

**Figure 5 diagnostics-15-02877-f005:**
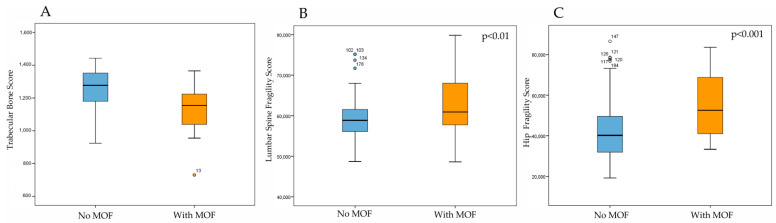
Trabecular Bone Score by DXA (**A**), Fragility Score at lumbar spine (**B**) and Fragility score at hip (**C**) by REMS technique in DM2 patients with or without MOF.

**Table 1 diagnostics-15-02877-t001:** Anthropometric, clinical and densitometric characteristics of the T2DM patients and the controls.

	T2DM Patients (*N* = 223)	Controls (*N* = 102)	*p*
M/F	97/126	45/57	n.s.
Age (yrs)	67.5 ± 9.1	68.7 ± 7.5	n.s.
Weight (Kg)	78.6 ± 15.7	74.1 ± 12.3	0.05
Height (cm)	164.0 ± 8.6	165.1 ± 6.7	n.s.
BMI (Kg/m^2^)	29.2 ± 5.1	27.6 ± 4.3	0.01
HbA1c (%)	6.8 ± 1.2	--------	
T2DM duration (yrs)	12.8 ± 10.6	---------	
Creatinine (mg/dL)	1.0 ± 0.3	0.9 ± 0.2	n.s.
Calcium (mg/dL)	9.4 ± 0.5	9.3 ± 0.4	n.s.
Phosphate (mg/dL)	3.6 ± 0.6	3.5 ± 0.6	n.s.
25OHD (ng/mL)	21.6 ± 11.3	23.8 ± 9.4	n.s.
PTH (pg/mL)	35.5 ± 16.7	33.6 ± 15.8	n.s.
DXA LS-BMD (g/cm^2^)	1.070 ± 0.211	0.946 ± 0.189	0.01
DXA FN-BMD (g/cm^2^)	0.792 ± 0.162	0.730 ± 0.178	0.05
DXA TH-BMD (g/cm^2^)	0.936 ± 0.157	0.887 ± 0.178	0.05
REMS LS-BMD (g/cm^2^)	0.871 ± 0.119	0.893 ± 0.120	0.05
REMS FN-BMD (g/cm^2^)	0.724 ± 0.120	0.733 ± 0.099	n.s.
REMS TH-BMD (g/cm^2^)	0.865 ± 0.170	0.872 ± 0.117	0.05

## Data Availability

The data that support the findings of this study are available from the corresponding author (C.C.).

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
