# Peer review of "Utility of REMS-Derived Fragility Score and Trabecular Bone Score in Evaluating Bone Health in Type 2 Diabetes Mellitus"

_diagnostics, 2025, doi:10.3390/diagnostics15222877_

Round 1
Reviewer 1 Report
Comments and Suggestions for Authors
Thank you for your effort on the study and the manuscript.
My comment:
The abstract section is well-written. There is no specific article on this topic in the literature as far as I know. So the article fills the gap on that topic.
REMS has not been used in diabetic patients in previous articles.
The introduction, materials, and results section of the article are sufficient. The first paragraph of the introduction section is too long, so it makes it hard to follow. Please divide this paragraph into 2 or 3 paragraphs.
Yet the discussion section is insufficient. In the discussion section, you should discuss your results with the literature. But you mostly give literature data, and only in one paragraph you discussed your results with the literature. This section should be rewritten.
The references, the tables, and the figures are appropriate. They do not need to be improved.
The materials and methods section and results sections are well-written
Author Response
The abstract section is well-written. There is no specific article on this topic in the literature as far as I know. So the article fills the gap on that topic.
REMS has not been used in diabetic patients in previous articles.
The introduction, materials, and results section of the article are sufficient. The first paragraph of the introduction section is too long, so it makes it hard to follow. Please divide this paragraph into 2 or 3 paragraphs.
Yet the discussion section is insufficient. In the discussion section, you should discuss your results with the literature. But you mostly give literature data, and only in one paragraph you discussed your results with the literature. This section should be rewritten.
The references, the tables, and the figures are appropriate. They do not need to be improved.
The materials and methods section and results sections are well-written
Following the reviewer's suggestion:
1) we have broken the introduction into several paragraphs
2) we have restructured and expanded the discussion section. Moreover, we have added 4 references
Reviewer 2 Report
Comments and Suggestions for Authors
Strengths
The objective is relevant: to compare TBS and REMS parameters (BMD and FS) in patients with T2DM.
Adequate sample size (223 T2DM vs. 102 controls) and detailed methodology.
Coherent results and well-structured discussion, contextualizing known limitations of TBS and the potential of REMS.
Simultaneous use of DXA, REMS, and TBS reinforces comparative validity.
Good presentation of the results, with an organized presentation, including explanatory tables and figures.
But...
Correct the small formatting error in the subsection title: "2.5. Statistical analysis" appears duplicated: 2.5. Fracture Assessment and 2.5. Statistical analysis.
Adequate sample size (223 T2DM vs. 102 controls) and detailed methodology. However, in Figure 1 we have: T2DM men N=97 and T2DM women N=136, total = 233 T2DM. Okay, which one would be true?!
Could you also indicate the software and version used for TBS ("TBS iNsight version 2.1, Medimaps SA, Bordeaux, France") in the Methods section?
Discussion: Avoid repetition between paragraphs 307–317 and 318–324 (both describe discrepancies between DXA and REMS).
Replace redundant expressions like "however, however" (line 231).
Regarding the text in Figure 1, you don't have the distribution of study participants, only the T2DM.
Author Response
Reviewer 2
Strengths
The objective is relevant: to compare TBS and REMS parameters (BMD and FS) in patients with T2DM.
Adequate sample size (223 T2DM vs. 102 controls) and detailed methodology.
Coherent results and well-structured discussion, contextualizing known limitations of TBS and the potential of REMS.
Simultaneous use of DXA, REMS, and TBS reinforces comparative validity.
Good presentation of the results, with an organized presentation, including explanatory tables and figures.
But...
Correct the small formatting error in the subsection title: "2.5. Statistical analysis" appears duplicated: 2.5. Fracture Assessment and 2.5. Statistical analysis.
According to the suggestion of the Reviewer The title of the Statistical analysis subsection has been corrected to “2.6. Statistical analysis.”
Adequate sample size (223 T2DM vs. 102 controls) and detailed methodology. However, in Figure 1 we have: T2DM men N=97 and T2DM women N=136, total = 233 T2DM. Okay, which one would be true?!
The number of patients with T2DM reported in the Abstract and Table 1 is 223 (97 men and 126 women); therefore, the correct number of women with T2DM in Figure 1 is 126 instead of 136
Could you also indicate the software and version used for TBS ("TBS iNsight version 2.1, Medimaps SA, Bordeaux, France") in the Methods section?
The version used is: “version 2.2.0.0, Medimaps SA, Bordeaux, France”
Discussion: Avoid repetition between paragraphs 307–317 and 318–324 (both describe discrepancies between DXA and REMS).
The Discussion section has been partially rewritten, and the previous repetition regarding the discrepancies between DXA and REMS has been removed.
Replace redundant expressions like "however, however" (line 231).
One of the two instances of “however” has been deleted
Regarding the text in Figure 1, you don't have the distribution of study participants, only the T2DM.
Following the referee’s suggestion, Figure 1 has been updated to include the distribution of the control group as well.
Reviewer 3 Report
Comments and Suggestions for Authors
The article is well structured, comprehensively covering the phenomena of bone fragility in diabetes, the advantages and limitations of different assessment methods, and highlighting the potential of REMS as a diagnostic tool.
The main novelty of the study lies in the evaluation of new diagnostic methods for bone fragility in patients with T2DM. In particular, it highlights the potential of REMS (radiofrequency ultrasound) technology, a non-invasive, radiation-free method that, through the fragility score (FS), can assess bone quality independently of BMD. Previous studies have shown that FS can discriminate more effectively between patients with and without fractures than traditional techniques, but until now there has been no data on the predictive value of FS for fractures in T2DM.
Suggestions for improvement:
o The legend should be specified under each figure.
o The yellow underlining should be deleted.
o A more in-depth discussion of the practical implications of using REMS in screening and monitoring.
For a complete approach, a more detailed discussion of how these results may influence clinical practice and future research directions would be useful.
Author Response
Reviewer 3
The article is well structured, comprehensively covering the phenomena of bone fragility in diabetes, the advantages and limitations of different assessment methods, and highlighting the potential of REMS as a diagnostic tool.
The main novelty of the study lies in the evaluation of new diagnostic methods for bone fragility in patients with T2DM. In particular, it highlights the potential of REMS (radiofrequency ultrasound) technology, a non-invasive, radiation-free method that, through the fragility score (FS), can assess bone quality independently of BMD. Previous studies have shown that FS can discriminate more effectively between patients with and without fractures than traditional techniques, but until now there has been no data on the predictive value of FS for fractures in T2DM.
Suggestions for improvement:
o The legend should be specified under each figure.
o The yellow underlining should be deleted.
o A more in-depth discussion of the practical implications of using REMS in screening and monitoring.
For a complete approach, a more detailed discussion of how these results may influence clinical practice and future research directions would be useful.
We thank the referee for the valuable suggestions. In particular:
- We have verified that each figure includes an appropriate legend.
- The yellow highlights have been removed.
- The Discussion section has been extensively revised, and the practical implications of using REMS for screening T2DM patients have been discussed and duly considered.
Reviewer 4 Report
Comments and Suggestions for Authors
Dear Authors
In the abstract, the statistical tests are missing.
The manuscript is well designed, the authors establish the study objective, and describe the methodology. Note that I asked to check the sample numbers. The results are presented in one table and four figures (the table should fit on one page). The discussion is adequate and highlights the study's limitations. The conclusion answers the study objective.
The sample described in the abstract differs from that in the Methods (although it meets the ones in figure 1). Please check the number of excluded (35 or 37?).
Regards
Author Response
Reviewer 4
Dear Authors
In the abstract, the statistical tests are missing.
The manuscript is well designed, the authors establish the study objective, and describe the methodology. Note that I asked to check the sample numbers. The results are presented in one table and four figures (the table should fit on one page). The discussion is adequate and highlights the study's limitations. The conclusion answers the study objective.
The sample described in the abstract differs from that in the Methods (although it meets the ones in figure 1). Please check the number of excluded (35 or 37?).
We thank the referee for the constructive comments.
As stated in the Abstract and shown in Table 1, a total of 223 diabetic patients were included in the study. We have reviewed the manuscript and corrected some typographical errors. In addition, following another reviewer’s suggestion, we have updated the flow chart (Figure 1) to include the number of controls.
We did not include the description of the statistical tests in the Abstract, as doing so would have made it excessively long